# Multiple CH/π Interactions Maintain the Binding of Aflatoxin B_1_ in the Active Cavity of Human Cytochrome P450 1A2

**DOI:** 10.3390/toxins11030158

**Published:** 2019-03-12

**Authors:** Jun Wu, Sisi Zhu, Yunbo Wu, Tianqing Jiang, Lingling Wang, Jun Jiang, Jikai Wen, Yiqun Deng

**Affiliations:** 1Guangdong Provincial Key Laboratory of Protein Function and Regulation in Agricultural Organisms, College of Life Sciences, South China Agricultural University, Guangzhou 510642, China; wujun@scau.edu.cn (J.W.); zhusisi@stu.scau.edu.cn (S.Z.); wuyunbo5@stu.scau.edu.cn (Y.W.); jiangtianqing@stu.scau.edu.cn (T.J.); llwang417@scau.edu.cn (L.W.); jiangjun@scau.edu.cn (J.J.); jkwen@scau.edu.cn (J.W.); 2Key Laboratory of Zoonosis of Ministry of Agriculture and Rural Affairs, South China Agricultural University, Guangzhou 510642, China

**Keywords:** CH/π interaction, cytochrome P450, CYP1A2, aflatoxin, specificity

## Abstract

Human cytochrome P450 1A2 (CYP1A2) is one of the key CYPs that activate aflatoxin B_1_ (AFB_1_), a notorious mycotoxin, into carcinogenic exo-8,9-epoxides (AFBO) in the liver. Although the structure of CYP1A2 is available, the mechanism of CYP1A2-specific binding to AFB_1_ has not been fully clarified. In this study, we used calculation biology to predict a model of CYP1A2 with AFB_1_, where Thr-124, Phe-125, Phe-226, and Phe-260 possibly participate in the specific binding. Site-directed mutagenesis was performed to construct mutants T124A, F125A, F226A, and F260A. *Escherichia coli*-expressed recombinant proteins T124A, F226A, and F260A had active structures, while F125A did not. This was evidenced by Fe^2+^∙Carbon monoxide (CO)-reduced difference spectra and circular dichroism spectroscopy. Mutant F125A was expressed in HEK293T cells. Steady kinetic assays showed that T124A had enhanced activity towards AFB_1_, while F125A, F226A, and F260A were significantly reduced in their ability to activate AFB_1_, implying that hydrogen bonds between Thr-124 and AFB_1_ were not important for substrate-specific binding, whereas Phe-125, Phe-226, and Phe-260 were essential for the process. The computation simulation and experimental results showed that the three key CH/π interactions between Phe-125, Phe-226, or Phe-260 and AFB_1_ collectively maintained the stable binding of AFB_1_ in the active cavity of CYP1A2.

## 1. Introduction

Human cytochrome P450 1A2 (CYP1A2) is an important CYP isoform expressed in the liver that broadly participates in the metabolism of numerous endogenous substrates and xenobiotics [1,2,3,4,5,6,7]. In vivo, CYP1A2 contributes largely to the metabolic clearance of drugs such as caffeine, theophylline, tacrine, tizanidine, etc. For in vitro studies, 7-ethoxyresorufin and 7-ethoxycoumarin are often used as the probe substrates to quantify CYP1A2 activity. CYP1A2 is apt at metabolizing diverse polynuclear aromatic hydrocarbons and plays important roles in the bioactivation of arylamines into carcinogenic compounds [8]. This bioactivation is also observed in the metabolism of aflatoxins by CYP1A2 [9].

Aflatoxin B_1_ (AFB_1_) is a ubiquitous contaminant found in a number of cereal crops both pre- and post-harvest, and it is of concern due to its toxic and carcinogenic properties [10]. The carcinogenic potency of AFB_1_ is closely related to the formation of highly reactive exo-8,9-epoxide (AFBO) [11,12]. CYP1A2 is one of the key P450 enzymes involved in the metabolism of AFB_1_ and plays important roles in AFB_1_ bioactivation [9,13,14]. Because of the broad substrate spectrum, it is necessary to understand the structure–function relationships underlying the catalytic mechanism of CYP1A2 using classic biochemical methods, such as site-directed mutagenesis and enzyme kinetic analysis. It has been reported that a triple mutant E163K/V193M/K170Q has a five-fold increase in activity compared to CYP1A2 wide-type (WT) in the metabolism of 7-methoxyresorufin [15]. The mutagenesis of Leu-382 affects substrate specificity in metabolizing a few of the drugs, including phenacetin, 7-ethoxyresorufin, and 7-methoxyresorufin [3,16,17]. Thr-124 and Phe-226 play key roles in the binding of mexiletine [1]. The crystal structure of CYP1A2 shows a series of residues arranged in the substrate binding cavity, among which Phe-226 is observed to form π–π stacking with α-naphthoflavone [18]. Interestingly, this interaction was found to be involved in the binding of flavonoid derivatives and 7-ethynylcoumarins [19,20].

Although the structural information of the substrate-binding pocket can be observed, the key mechanism of CYP1A2 when specifically recognizing and binding to AFB_1_ has not been fully clarified. Herein, using computational simulations and experimental verifications, we investigated the relationships between CYP1A2’s structure and activity relating to AFB_1_ activation, and we determined CYP1A2’s key residues responsible for binding to AFB_1_. More importantly, we demonstrated that the CH/π interaction governed the specific binding of CYP1A2 with AFB_1_. This information is essential for understanding how CYP1A2 recognizes and binds to AFB_1_.

## 2. Results and Discussion

### 2.1. Molecular Dynamics Simulation

To study the mechanism of CYP1A2 in recognizing and binding to AFB_1_, we performed molecular docking simulations to obtain the active binding mode in the catalytic pocket of CYP1A2. A thousand docking conformations of CYP1A2 with AFB_1_ were obtained, among which 10 conformations with lower docking energy were retained. The one with lowest energy (−8.98 kcal/mol, Appendix A) was selected as the best and was analyzed. The best docking pose was further evaluated using molecular dynamics (MD) simulations. Following 100 ns of MD simulations, the trajectory was extracted and analyzed for detailed information on the receptor–ligand interaction.

The time-dependent root mean square deviations (RMSD) for the 100 ns production-phase MD were calculated and plotted, as shown in Figure 1A. As can be seen, the system reached equilibrium at about 50 ns and stayed stable until the end of the 100 ns simulation. The low RMSD values of the protein backbone, the binding pocket, and the ligand indicated no dramatic conformational changes during the whole process (i.e., close to the crystal structure), thus indicating a high confidence level for results obtained from this analysis.

The distances between AFB_1_ and heme were also measured, especially for the closest C9 or C8 atoms (Figure 1B) of AFB_1_ to the Fe atom of heme. Compared to C9, C8 is closer to the Fe center of heme, which is from about 4–5 Å during the 100 ns MD simulation. According to the co-crystal structure, the shortest interatomic distance between the co-crystal ligand to heme is 4.2 Å, which conforms to the distance requirement estimated by the accessibility criteria for C-oxidation [21] and allows for the contact of the ferric peroxy anion and the following transfer of the oxygen atom mediated by Cpd I [22]. Meanwhile, the MD simulation produced the dominant AFB_1_ conformation that was favorable for the formation of AFBO (Appendix A).

To better visualize the receptor–ligand binding mode, we also did conformational clustering on the trajectory and extracted the most representative frame out of the largest cluster. As shown in Figure 2, AFB_1_ forms multiple interactions with the receptor, including a hydrogen bond with Thr-124 and stackings with Phe-125, Phe-226, and Phe-260. Three CH/π interactions were observed between the side-chain benzene rings of Phe-125, Phe-226, and Phe-260 and the B ring, the A ring (3β-) and (2α-) hydrogen atom in AFB_1_, respectively (Figure 2). There was also a water-mediated hydrogen bond with Thr-118 and Asp-313 (Figure 2). The time dependency of these interactions was also calculated and plotted as Figure 1C–G, respectively, which were relatively stable once equilibrium was achieved.

To further verify the binding model, we also performed the docking of AFB_1_ to the mutants T124A, F125A, F226A, and F260A, respectively. The results showed that the binding energy between AFB_1_ and the mutant was low (<8 kcal/mol, Appendix A). Interestingly, it was also found that C8 and C9 of AFB_1_ were not oriented towards the heme iron in all poses, which meant there was inactive conformation and no AFBO was able to form (Appendix A).

### 2.2. The Spectral Property of Escherichia coli-Expressed CYP Proteins

To evaluate the structural features of the *E. coli*-expressed CYP proteins, we performed Fe^2+^·CO versus Fe^2+^ difference spectra and circular dichroism (CD) spectroscopy. As shown in Figure 3A, the recombinant CYP1A2 wide-type (WT), T124A, F226A, and F260A had a strong absorbance peak at around 450 nm, and the absorbance peaks at 420 nm were weak or did not exist. For F125A, we could not obtain a protein with a 450 nm absorbance peak; instead we found one with a 425 nm peak (Appendix A). Phe-125 was also replaced by Val (Appendix A). These results showed that the *E. coli*-expressed CYP1A2, T124A, F226A, and F260A possessed the typical spectral characteristics of functional P450. Furthermore, the far-UV CD spectra of these proteins displayed two obvious minus peaks located at 208 and 222 nm, respectively (Figure 3B), implying the representative secondary structures were rich in α-helixes [24]. The contents of the secondary structure of the proteins were calculated (Appendix A).

### 2.3. Activity Analysis

To further verify whether the amino acid sites predicted by calculation simulation were involved in the specific binding of AFB_1_ in active pockets, we examined the catalytic activities of CYP1A2 and the mutants towards AFB_1_. The enzymatic kinetics that CYP1A2 WT, T124A, and F260A used to oxidize AFB_1_ into the major metabolic product AFBO conformed to the Michaelis–Menten equation (Figure 3C). The apparent *K*_m_ and *K*_cat_ calculated by nonlinear regression of WT were 29.86 ± 2.17 μM and 0.24 ± 0.007 min^−1^ (Table 1), respectively. For mutant T124A, the kinetic parameters *K*_m_ notably decreased to 6.79 ± 0.79 μM (Table 1), while *K*_cat_ remained almost unchanged (0.20 ± 0.004 min^−1^, Table 1), leading to the obvious enhancement of the catalytic efficiency *K*_cat_/*K*_m_ (3.69-fold) compared to the WT. The preceding MD simulation has showed that Thr-124 was able to form hydrogen bonds with AFB_1_ (Figure 2). Since the mutant resulted in the loss of hydrogen bonds in T124A and this did not influence the enzyme activity for AFBO formation, we can draw a conclusion that Thr-124 does not contribute to the specific binding of AFB_1_. By contrast, the substitution of Phe-226 with Ala totally abolished AFB_1_ metabolism, with no AFBO-glutathione (GSH) being detected (Table 1). This implies a strong correlation between the structure and the activity at this site. In the case of F125A, because of the abnormal CO-reduced spectrum, this mutant was expressed in human HEK293T cells (Figure 3D). The incubation of S9 fractions with AFB_1_ showed that the eukaryotic cell-expressed CYP1A2 WT was able to metabolize AFB_1_ into AFBO-GSH (Figure 3F). Meanwhile, similar to the negative control (Figure 3E), no product peak appeared for F125A (Figure 3G). This result suggests that the replacement of Phe-125 with Ala also resulted in its inactivation in AFBO generation. For F260A, although the metabolite AFBO-GSH was detected, the apparent *K*_m_ and *K*_cat_ were only 457.35 ± 387.66 μM and 0.096 ± 0.067 min^–1^, respectively (Table 1), which resulted in quite a low catalytic efficiency (*K*_cat_/*K*_m_) (only 2.5% of WT, Table 1), exhibiting extremely weak metabolic capacity.

It should be noted that, despite the docking simulation of the mutants showing that Thr-124, Phe-125, Phe-226, and Phe-260 all participated in the binding to AFB_1_ (Appendix A), the activity profiles of the four alanine mutants examined reflected their authentic correlation to AFB_1_ metabolism. Thr-124 interacted with AFB_1_ through the hydrogen bond in this study, whereas it was not what determined the extent of the activity, judging from the activity result. In contrast, this site contributed to the metabolism of mexiletine by bridging a water molecule (WAT) [1]. A molecular docking study showed that AFB_1_ binds to CYP1A2 using its A ring’s aryl–alkyl interactions with Phe-226 and Phe-260 [21]. In the present study, based on the results of calculation simulation and activity assays, we guessed that Phe-125, Phe-226, and Phe-260 were all involved in AFB_1_’s binding to a great degree through key CH/π interactions. Because most of the substrates metabolized by CYP1A2 are hydrophobic aromatic ring compounds, it is expected that bulky aromatic residues like phenylalanine participate in substrate binding. It has been reported that Phe-226 plays an important role in the binding of multiple drugs, and usually forms π–π stacking with the drug molecules in the active cavity [1,18,19,20]. Namely, the aromatic rings adopt a mutually parallel orientation in the interaction. In the metabolism of 7-ethoxyresorufin and phenacetin, this residue plays an equally important role [26]. Despite Phe-226 being mutated into isoleucine, threonine, or tyrosine, its activity remains low [26]. Phe-125 contributes to a tight binding affinity to α-naphthoflavone through an orthogonal aromatic interaction [18], which was proven to be a π–π interaction (T-Shaped) between a flavone ring and the phenyl side chain of Phe-125 [20]. Different from π–π stacking, the CH/π interaction is a mutual attraction between the C–H bond and the π system [27], which is a dispersion force in nature [28,29] and has been demonstrated to play enormously important roles in protein stability, protein–protein interactions, biomolecular recognition, and substrate specificity in an increasing number of studies [30,31,32,33,34,35,36]. A detailed study that investigates 1154 protein structures from the Protein Data Bank (PDB) claims that approximately 75% of Trp, 50% of Phe and Tyr, and 25% of His are involved in CH/π interactions [37]. Although the CH/π interaction is also thought to be a kind of weak hydrogen bond (with a bond energy of 1.5–2.5 kcal/mol [30]), it maintained the stable binding of AFB_1_ in a substrate pocket of CYP1A2. The three CH/π interactions that phenylalanines participate in are important, and it is speculated that AFB_1_ tends to lose the correct conformation that facilitates AFBO formation in the absence of any of them.

## 3. Conclusions

In summary, this study revealed that Phe-125, Phe-226, and Phe-260 collectively played critical roles in AFB_1_-specific binding in the cavity of CYP1A2. This binding was maintained by respective key CH/π interactions that each phenylalanine participated in. These interactions are indispensable and keep AFB_1_ in the correct conformation that facilitates the formation of AFBO. This reflects the roles of CH/π interaction in determining AFB_1_ binding with CYP1A2, which aids in the bioactivation of AFB_1_ by CYP1A2. In the meantime, differing from other studies, a single CH/π interaction is not sufficient to maintain the stable binding of AFB_1_ in our study. As a result, there needs three CH/π interactions to collectively preserve the active conformation of AFB_1_ for AFBO formation. The conservation of phenylalanine residues with π systems is linked to the function of CYP1A2. This study provides further insights into the structural basis of CYP1A2 in bioactivating AFB_1_ into carcinogenic AFBO. 

## 4. Materials and Methods

### 4.1. Materials

AFB_1_ standard was purchased from Qingdao Pribolab Bioengineering Limited Corporation (Qingdao, China). Restriction endonucleases and high-fidelity DNA polymerase were purchased from TaKaRa Biomedical Technology Corporation (Beijing, China). A QuikChange site-directed mutagenesis kit was purchased from Stratagene (Guangzhou, China). δ-aminolevulinic acid was purchased from Sigma Aldrich (St. Louis, MO, USA). Isopropyl-β-D-thiogalactoside (IPTG), NADPH, and BCA protein quantitation kits were purchased from the Beijing Dingguo Changsheng Biotechnology Limited Corporation. A 5 mL HisTrap HP column was purchased from GE healthcare (Uppsala, Sweden). A ZORBAX SB-C18 column (3.0 × 100 mm, 1.8 microns) was purchased from Agilent (Santa Clara, CA, USA). HEK293T cells (CRL-3216) were purchased from ATCC. Chromatographic grade acetonitrile, methanol, and formic acid were purchased from Thermo Fisher Scientific (Rockford, IL, USA). All primers were synthesized in Invitrogen (Guangzhou, Guangdong province, China). All the other reagents and chemicals used were of the highest analytical grade available.

### 4.2. Molecular Docking

The 3D structure of AFB_1_ was built in Molecular Operating Environment (MOE) v2015.1001. The protein structure of CYP1A2 was downloaded from the RCSB Protein Data Bank (PDB code: 2HI4) [38] and prepared with the default structure preparation workflow in MOE to fix issues like missing atoms and/or non-standard atom names. 

The protonation state of the protein and the orientation of the hydrogens were optimized by LigX, at a pH of 7 and temperature of 300 K. An AMBER10:EHT force field and the implicit solvation model of the reaction field (R-field) were applied prior to docking. An MOE-dock was used for docking AFB_1_ to CYP1A2 following the “induced fit” protocol, in which the side chains of the receptor pocket were allowed to move according to ligand conformations, with a constraint on their positions. The weight used for tethering side chain atoms to their original positions was set to 10. The docked poses were ranked by London dG scoring first, then a force field refinement was carried out on the top scoring poses, followed by a rescoring of GBVI/WSA dG. 

The receptor–ligand complex was then neutralized by adding sodium or chlorine counter ions, and solvated in a cuboid box of TIP3P [39] water molecules with solvent layers 10 Å between the box edges and solute surface. The ligand atoms were optimized using the Gaussian09 [40] package at the level of HF/6–31 g*, then the partial atomic charges were calculated using the restrained electrostatic potential (RESP) [41] charge from the calculation with the Gaussian09 package at the HF/6–31 g* level. 

### 4.3. MD Simulation

All MD simulations were performed using AMBER16 [42,43]. The AMBER GAFF and AMBER14SB force fields were applied, and the SHAKE [44] algorithm was used to restrict all covalent bonds involving hydrogen atoms with a time step of 2 fs. The particle-mesh Ewald (PME) method [45] was employed to treat long-range electrostatic interactions. For each solvated system, two steps of minimization were performed before the heating step. The first 4000 cycles of minimization were performed with all heavy atoms restrained with 50 kcal/(mol·Å^2^), whereas solvent molecules and hydrogen atoms were given freedom to move. Then, non-restrained minimization was carried out involving 2000 cycles of steepest descent minimization and 2000 cycles of conjugated gradient minimization. Afterwards, the whole system was heated from 0 to 300 K in 50 ps using Langevin dynamics at a constant volume. Then, it was equilibrated for 400 ps at a constant pressure of 1 atm. A weak constraint of 10 kcal/ (mol·Å^2^) was used to restrain all the heavy atoms during the heating steps. Periodic boundary dynamic simulations were carried out for the whole system with a constant composition, pressure, and temperature (NPT) ensemble at a constant pressure of 1 atm and 300 K in the production step. In the production phase, a 100 ns simulation was carried out. The trajectories were further analyzed using the CPPTRAJ module of AMBER16.

### 4.4. Vector Construction and Site-Directed Mutagenesis

The human CYP1A2 gene was fully synthesized with optimized codon usage bias for *E. coli*. To improve soluble expression, bovine 17α leader peptide was introduced to the N-terminus of CYP1A2, with a 6×-His Tag sequence appended to the C-terminus [46]. The modified sequence was ligated into the pCW *Ori*^+^ expression vector. The resulting pCW/1A2 plasmid was used as the DNA template for oligonucleotide-directed mutagenesis. The T124 A, F125A, F226A, and F260A mutants were prepared using a QuikChange site-directed mutagenesis kit. The primers used for mutagenesis were 5′-CGATGGTCAGTCTCTGGCGTTCAGCACTGACAGCG-3′ for T124A, 5′-GATGGTCAGTCTCTGACCGCAAGCACTGACAGCGGCCCG-3′ for F125A, 5′-GAAGAACACCCACGAGGCGGTGGAGACTGCTTCTT-3′ for F226A, and 5′-CAAAGCCTTTAACCAGCGCGCCCTGTGGTTCCTGCAGAAGAC-3′ for F260A. All of the mutations were verified using DNA sequencing.

### 4.5. Protein Expression and Purification

The recombinant human NADPH P450 reductase was expressed and purified as previously described [47]. The recombinant plasmids containing CYP1A2 cDNA, T124A, F226A, and F260A were transformed into DH5α for large-scale protein expression. For the F125A mutant, because the CO-reduced difference spectrum of the expressed protein had no typical 450 nm absorbance peak, we used a mammalian cell expressing system to obtain the S9 fraction. *E. coli* membranes containing recombinant proteins were prepared as described earlier [48]. Briefly, membranes were gently stirred and solubilized for 2 h at 4 °C in a 100 mM potassium phosphate buffer (with a pH of 7.4) containing 20% glycerol, 1 mM EDTA, 500 mM KCl, 1% CHAPS, and 20 mM imidazole. The suspension was centrifuged and filtered through a 0.45 μm nylon membrane before being loaded onto a 5 mL HisTrap HP column that had been pre-equilibrated with a 100 mM potassium phosphate buffer (pH 7.4) containing 20% glycerol, 1 mM EDTA, 500 mM KCl, 0.5% CHAPS, and 20 mM imidazole. The recombinant proteins were eluted with a gradient program of 100–500 mM imidazole. The fractions containing the protein of interest were combined and dialyzed at 4 °C for 36 h. Aliquots of proteins were stored at −80 °C. 

### 4.6. Cell Assays

HEK293T cells were digested using trypsin and grown in DMEM medium containing 10% fetal calf serum in 10 cm culture plates in 5% CO_2_ atmosphere at 37 °C overnight. The open reading frame (ORF) region of CYP1A2 WT and F125A were constructed into a pcDNA3.1/myc-His(-)A vector, respectively (Thermo Fisher Scientific, USA). The negative control pcDNA3.1 empty vector and expression plasmids pcDNA/CYP1A2 and pcDNA/F125A were transiently transfected into HEK293T cells, respectively. After being cultured at 37 °C for 24 h, the cells were harvested and resuspended in a 500 μL potassium phosphate buffer (100 mM, pH 7.4), then sonicated on ice for 10 cycles at 10% amplitude for 5 s at 10-s intervals. The cell homogenate was centrifuged at 9000× *g* for 20 min at 4 °C, and the supernatant was the S9 fractions. 

### 4.7. Immunoblotting Analysis

Proteins from S9 fractions were separated by 10% SDS-PAGE, then electrophoretically transferred onto a PVDF membrane (PALL, Ann Arbor, MI). The membrane was blocked and incubated with an anti-Myc-tag primary antibody (#2276, Cell Signaling Technology, 1:1000 dilutions, Danvers, MA, USA). The bands were detected using the BeyoECL Star chemiluminescence kit (Beyotime Biotechnology, Shanghai, China).

### 4.8. CO Difference Spectra

Fe^2+^∙CO versus Fe^2+^ difference spectra of the recombinant CYP1A2 and the mutants were measured with a UV-2550 spectrophotometer (SHIMADZU, Kyoto, Japan). 200 μL of the purified proteins was suspended in 1.8 mL of Tris·HCl buffer (containing 50 mM, 20% glycerol, 1.25% CHAPS, and 1 mM EDTA at a pH of 7.4) and divided into two cuvettes. A baseline spectrum from 500 to 400 nm was recorded. CO was bubbled slowly into the sample cuvette for 30 s, then a few drops of dithionite was added into the sample cuvette, where the solution was pipetted and placed for 1 min until the difference spectrum was recorded. The concentrations of active CYPs were calculated using the extinction coefficient ε_450-490_ = 91 mM^−1^ cm^−1^ [49].

### 4.9. Circular Dichroism Spectroscopy

The far-UV circular dichroism (CD) spectra of recombinant proteins were collected on a Chirascan spectrometer (Applied Photophysics, Leatherhead, UK) in the wavelength range of 200–260 nm, with a step size of 1 nm, a bandwidth of 0.5 nm, and 0.5 s collection time per step. The proteins were diluted into 50 mM potassium phosphate (5% glycerol, at a pH of 7.4). The concentrations of proteins were determined using the method of guanidine HCl denaturation [50]. The CD spectra were recorded at 4 °C, using a 0.1 cm quartz cuvette. The final spectra were the average of at least three repeats. The background spectra were collected and subtracted as described previously [51]. Finally, the corrected spectral data (obtained in millidegrees) were converted to mean residue molar ellipticities according to the protein concentrations [50].

### 4.10. Activity Assays

The standard incubation system consisted of 0.2 μM *E. coli*-expressed proteins, 0.8 μM NADPH cytochrome P450 reductase, 0.02 mg/mL L-α-dilauroyl-sn-glycero-3-phosphocholine, 3 mM glutathione, 16 μM mouse glutathione S-transferase (GST), AFB_1_ (concentrations ranged from 2 to 128 μM), 1 mM NADPH, and ultrapure water. This achieved a total volume of 200 μL. For eukaryotic cell-expressed proteins, the protein concentrations in S9 fractions were quantified using the BCA kit. The final concentration of S9 proteins (i.e., 4 mg/mL) from cells transfected with pcDNA3.1, CYP1A2 WT, and F125A were incubated with AFB_1_ using the system described above. The incubation reactions were performed for 30 min at 37 °C in a shaking bath and were terminated by adding 200 μL of ice-cold methanol. The solutions were kept at −20 °C overnight to facilitate protein precipitation, then centrifuged at 12,000× *g* for 10 min at 4 °C. The supernatants were filtered through a 0.22-μm nylon membrane before being injected into HPLC. The AFBO-GSH standard was prepared by incubating pig liver microsomes with AFB_1_ and GSH. Generally, aliquots of incubation containing 100 mM potassium phosphate (with a pH of 7.4), 0.88 mg/mL pig liver microsomes, 200 μM AFB_1_, 3 mM GSH, 16 μM mouse glutathione S-transferase, and 1mM NADPH were performed at 37 °C for 4 h. Next, the reactions were terminated with the addition of an equal volume of methanol. The samples were kept at −20 °C overnight before being centrifuged at 4 °C at 10,000g for 10 min. The supernatant was filtered and separated on a ZORBAX SB-C18 column. The mobile phases and the elution programs for separating AFBO-GSH and AFB_1_ were identical to what was described in the report [52]. The peak at 14.2 min for each separation was collected, merged, and dried under a stream of high purity nitrogen. The residues were dissolved with a 800 μL mixture of methanol (50%, *v*/*v*) and 100 mM potassium phosphate (50%, *v*/*v*) and separated on HPLC. Besides the product peak, there were also other impurity peaks. Therefore, the process of specific peak collection, dryness, dissolving, and separation was repeated until a single peak at 14.2 min remained. The final sample was identified by UV absorbance spectra to have a single maximum absorbance peak at 362 nm. Furthermore, it was identified as AFBO-GSH by LC–MS/MS–MS (Appendix A), which is consistent with the published research [53]. The concentration of AFBO-GSH was determined (1.33 μM, total volume 1.5 mL) by UV spectroscopy using a ɛ_362_ value of 21800 M^−1^cm^−1^ [32] and was used to establish a standard curve for HPLC quantitative analysis.

### 4.11. HPLC Assays

HPLC was performed on a Waters Alliance e2695 liquid chromatography system (Waters Corporation, Milford, MA, USA) equipped with a 2475 UV detector. Samples (100 µL) were separated at 25 °C on a ZORBAX SB-C18 column. The mobile phases and elution programs were as previously described [52]. The chromatographic separation was performed at a flow rate of 1 mL/min, and the absorbance was monitored at 362 nm.

### 4.12. Data Analysis

Because the reactions catalyzed by human CYP1A2 conform to classic Michaelis–Menten kinetics [54,55], the enzymatic reaction data were fitted to the Michaelis–Menten equation [56,57]:(1)$$v\;=\;\frac{V_{\max\;}\times \;\left[S\right]}{K_{m}+\left[S\right]}$$
where *v* and [*S*] are the reaction velocity and substrate concentration, respectively. *V*_max_ represents the maximum velocity. *K*_m_ is the Michaelis constant. The Michaelis–Menten curve fittings were performed using OriginPro 2017 (b9.4.0.220, OriginLab Corporation, Northampton, MA, USA), which generated the kinetic parameters *K*_m_ and *V*_max_. The turnover number (*k*_cat_) was calculated with *V*_max_ divided by the total enzyme concentration. The resulting data were expressed as the mean ± standard error of the mean (SEM) based on at least three independent experiments.

## Figures and Tables

**Figure 1 toxins-11-00158-f001:**
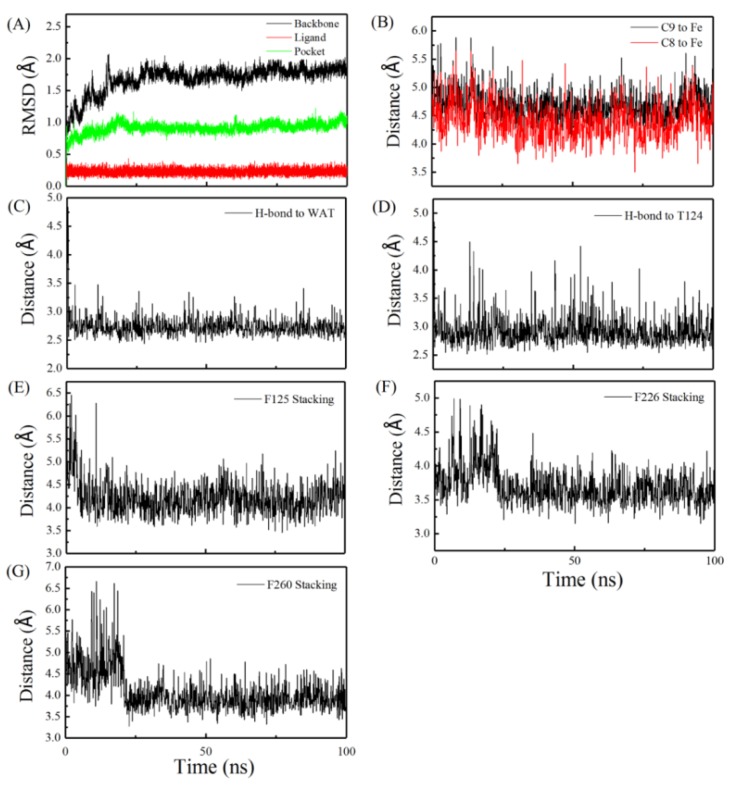
Trajectory analysis for the 100 ns MD of the solvated complex system of AFB_1_–CYP1A2. (**A**) Trajectory curves of the CYP1A2 protein backbone, AFB_1_, and pocket. (**B**) Trajectory curves of the distance of C8 and C9 to Fe. (**C**) Trajectory curves of the distance of the hydrogen bond to water. (**D**) Trajectory curves of the distance of the hydrogen bond between AFB_1_ and Thr-124. (**E**) Trajectory curves of the distance of the stacking interaction between AFB_1_ and Phe-125, (**F**) Phe-226, and (**G**) Phe-260.

**Figure 2 toxins-11-00158-f002:**
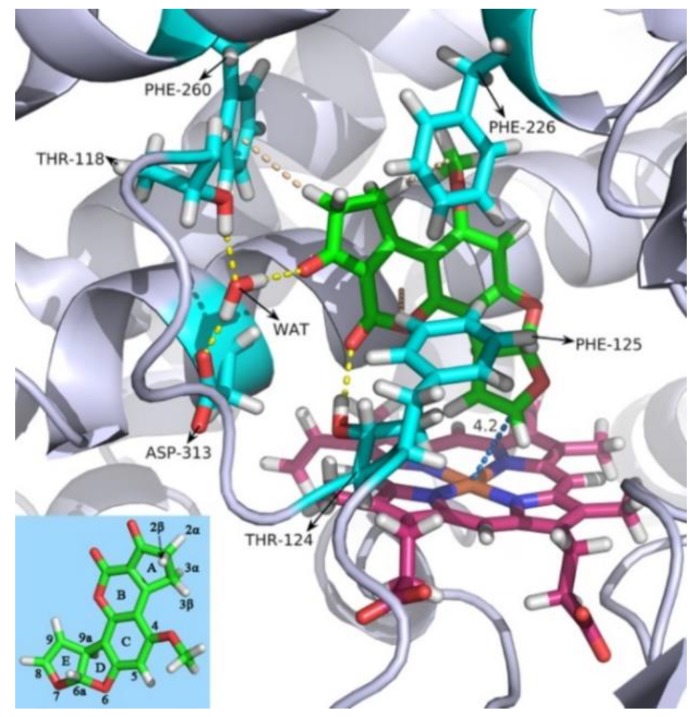
The binding mode of AFB_1_ with human CYP1A2 from the 100 ns MD simulation. Protein backbones are represented as a blue-white ribbon, and ferriporphyrin is represented as warm pink sticks. AFB_1_ is represented as green sticks, and the interactive amino acid residues are represented as cyan sticks. WAT: water molecule. Color scheme: red for oxygen atoms, blue for nitrogen atoms, and white for hydrogen atoms. Hydrogen bonds are represented as yellow dashed lines, and stacking interactions between AFB_1_ and the residues are represented as wheat dashed lines. The distances between atoms are represented as marine dashed lines and given in angstroms. The three-dimensional structure and atom numbering of AFB_1_ is shown in the lower left corner. Residues 107–117 are not shown, for clarity. This figure was rendered with PyMOL [23].

**Figure 3 toxins-11-00158-f003:**
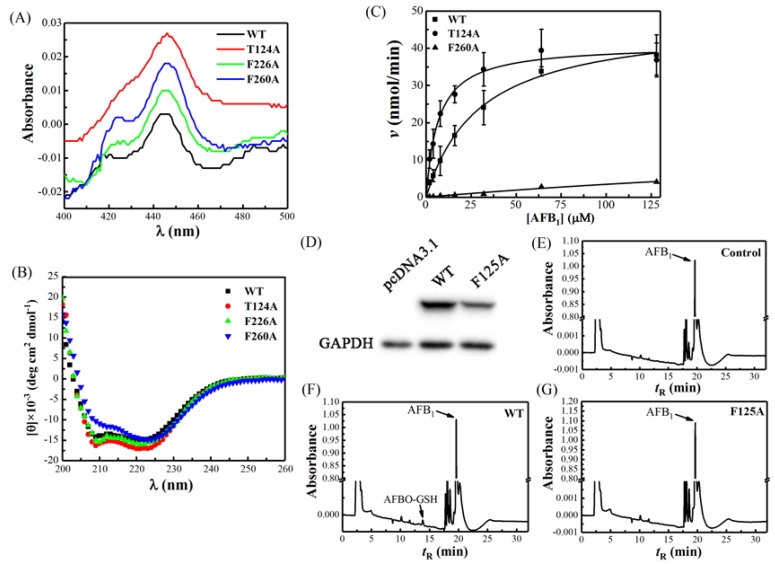
The spectral characteristics and the AFB_1_ metabolic activities of human CYP1A2 and the mutants. (**A**) Fe^2+^∙CO versus Fe^2+^ difference spectra of CYP1A2 WT, T124A, F226A, and F260A. The spectra were recorded at 25 °C and in 100 mM Tris∙HCl buffer (pH of 7.4) containing 10 mM CHAPS, 20% glycerol, and 1 mM EDTA. (**B**) Far-UV circular dichroism (CD) spectra of CYP1A2 WT, T124A, F226A, and F260A. CD spectra were recorded as before [25]. (**C**) The Michaelis–Menten equation fitting of *E. coli*-expressed CYP1A2, T124A, and F260A metabolizing AFB_1_ into AFBO-GSH. The data were analyzed by OriginPro 2017. (**D**) CYP1A2 WT and F125A were identified to efficiently express in HEK293T cells by Western blotting. High-performance liquid chromatography (HPLC) detection of metabolites from S9 incubation samples of (**E**) pcDNA3.1, (**F**) CYP1A2 WT, and (**G**) F125A, respectively.

**Table 1 toxins-11-00158-t001:** The steady-state kinetic parameters of CYP1A2 and the mutants oxidizing AFB_1_ into AFBO-GSH.

Proteins	*K*_m_ (μM)	*K*_cat_ (min^−1^)	*K*_cat_/*K*_m_ (min^−1^·mM^−1^)	Normalized Values (*K*_cat_/*K*_m_)
WT	29.86 ± 2.17	0.24 ± 0.007	8.0	1
T124A	6.79 ± 0.79	0.20 ± 0.004	29.5	3.69
F125A	ND	ND	ND	ND
F226A	ND	ND	ND	ND
F260A	457.35 ± 387.66	0.096 ± 0.067	0.2	0.025

Data are mean ± SEM based on at least three independent experiments. ND: not detected. WT: wide-type.

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
