# Peer review of "Multiple CH/π Interactions Maintain the Binding of Aflatoxin B1 in the Active Cavity of Human Cytochrome P450 1A2"

_toxins, 2019, doi:10.3390/toxins11030158_

Round 1

Reviewer 1 Report

My comments are also written in the attached file.

In this manuscript, amino acid residues which are important for binding to aflatoxin B1 were studied by mutagenesis analysis. The results are clear and I can accept this manuscript after minor modification. My comments are as follows.

1. Cytochrome P450 enzymes are very famous metabolic enzymes, and many studies have been performed about them. The studies about the amino acid residues which are important for binding to other chemicals should be cited in “Introduction”, and the results should be compared to yours in “Discussion”.

2. “5.10. Activity Assays”

The description about the method of AFGO-GSH preparation should be written more detailedly. In this study, the quality of the standard is very important. How much aflatoxin B1 did you use in the reaction of pig liver microsomes? The HPLC conditions for AFGO-GSH purification, the yield and the purity should be described in the text.

3. The binding ability of the mutant CYP1A2 to aflatoxin G1 is very interesting. Do you have the result?

Author Response

Point 1: Cytochrome P450 enzymes are very famous metabolic enzymes, and many studies have been performed about them. The studies about the amino acid residues which are important for binding to other chemicals should be cited in “Introduction”, and the results should be compared to yours in “Discussion”.

 Response 1: Thank you for the kind suggestion. We have followed the suggestion to supplement the related contents in “Introduction” and “Results and Discussion” in the revised version. Please see the highlighting parts in the revised version.

 Point 2: “5.10. Activity Assays” The description about the method of AFGO-GSH preparation should be written more detailedly. In this study, the quality of the standard is very important. How much aflatoxin B1 did you use in the reaction of pig liver microsomes? The HPLC conditions for AFGO-GSH purification, the yield and the purity should be described in the text. 

 Response 2: Thank you for the kind suggestion. Indeed, the quality of the standard AFGO-GSH is very important to this study. We have added the detailed method of AFBO-GSH preparation, HPLC conditions for AFBO-GSH purification, the yield and the purity in the revised version. We used 200uM AFB1 in a reaction of pig liver microsomes.

 Point 3: The binding ability of the mutant CYP1A2 to aflatoxin G1 is very interesting. Do you have the result? 

 Response 3: Thank you for providing the information. Frankly, we have no the result of the interaction between mutant CYP1A2 and aflatoxin B1 by far, since we focus on the metabolism of aflatoxin B1, the most toxic type, by cytochrome P450, not involved in the research on other aflatoxins including AFG1 yet. We will pay attention to the progress on this direction.

Reviewer 2 Report

The manuscript focused on the importance of CH/pi interaction of CYP1A2 specific binding to AFB1. However, the studies are good motivated, the manuscript may not be accepted for publication in the present form because there are some major concerns, which points need to be elucidated.

-          The descriptions of the applied methods are incomplete, e.g. the types of the spectrophotometer or CD spectrometer are missing. In Section 5. instead of the descriptions of the measurements several times just earlier publications were referred, however, at least a brief description of the applied methods and setups should be inserted. Furthermore, e.g. the indicated reference in section 5.8 also does not include the related measurement process.

-          The Michalis-Menten equation has been used but the equation and the related explanation of its applicability are completely missing. Furthermore, the related Km and Kcat parameters have to be also defined clearly.

-          The HF/6-31G* is not capable to describe CH/pi interaction.

-          The reliability of the docking simulation cannot be determined from the described results. Same basic information, such us, the numbers of the docking conformations, the related energies and structures have to be also inserted and analyzed. Furthermore, the binding of AFB1 to CYP1A2 mutants have to be also calculated with the aim to compare the data with the experimental results.

-          The complete analyses of the circular dichroism experiments are missing. I suggest inserting a table at least in the supplementary materials about the contents of the secondary structure elements of CYP1A2 and its mutants.

-          In the present form Section 3. reproduce number of information from the previous Section 2., therefore, it is suggested to rewrite this section or to write a combined Results and Discussion section.

-          The quality of Figure 3. have to be improved.

Author Response

Point 1: The descriptions of the applied methods are incomplete, e.g. the types of the spectrophotometer or CD spectrometer are missing. In Section 5. instead of the descriptions of the measurements several times just earlier publications were referred, however, at least a brief description of the applied methods and setups should be inserted. Furthermore, e.g. the indicated reference in section 5.8 also does not include the related measurement process.

Response 1: Thank you for the kind suggestion. Indeed, the manuscript lacks the detailed description of the applied methods. We have supplemented the contents about CO reduced difference spectra and CD spectroscopy. In addition, as you have pointed out, because the indicated reference in section 5.8 also does not include the related measurement process, we have deleted this citation here.

Point 2: The Michalis-Menten equation has been used but the equation and the related explanation of its applicability are completely missing. Furthermore, the related Km and Kcat parameters have to be also defined clearly.

Response 2: Thank you for the comments. We have supplemented the Michaelis-Menten equation and the explanation of its applicability, and defined the related Km and kcat. Meanwhile, four references have been cited here.

Point 3: The HF/6-31G* is not capable to describe CH/pi interaction.

Response 3: Indeed, the HF/6-31G* is not capable to describe CH/pi interaction. Actually, this method was used to perform ligand optimization prior to molecular dynamics in our study. And we used Molecular Mechanics force fields to analyze all interactions between the ligand and the receptor during MD and found the CH/pi interactions at specific sites.

Point 4: The reliability of the docking simulation cannot be determined from the described results. Same basic information, such us, the numbers of the docking conformations, the related energies and structures have to be also inserted and analyzed. Furthermore, the binding of AFB1 to CYP1A2 mutants have to be also calculated with the aim to compare the data with the experimental results.

Response 4: Thank you for the kind suggestion. We have supplemented the information of the dock simulation including docking conformations, the related energies and structures. Actually, we have performed the dockings of AFB1 to CYP1A2 mutants previously and these results have been provided in the supplemental material. Moreover, we discussed the data with the experimental results. Please see the highlighting parts in the revised version. Thanks!

Point 5: The complete analyses of the circular dichroism experiments are missing. I suggest inserting a table at least in the supplementary materials about the contents of the secondary structure elements of CYP1A2 and its mutants.

Response 5: Thank you for the kind suggestion. We have followed the suggestion and inserted a table about the contents of the secondary structure elements of CYP1A2 and its mutants in the supplementary materials.

Point 6: In the present form Section 3. reproduce number of information from the previous Section 2., therefore, it is suggested to rewrite this section or to write a combined Results and Discussion section.

Response 6: Thank you for the kind suggestion. For the sake of clarity, we have deleted the repeating information and wrote a combined Results and Discussion section.

Point 7: The quality of Figure 3. have to be improved.

Response 7: Thank you for the kind suggestion. We have redrawn Figure 3 to improve the quality.

Round 2

Reviewer 2 Report

The manuscript may be accepted for publication in the present form.

Author Response

We have revised the munascript again, in which Introduction was edited and two references were inserted. Please see the second revision of the manuscript.

Thank you!